# The Impact of a Severe El Niño Event on Vascular Epiphytes in Lowland Panama

Helena J. R. Einzmann [1,*], Letizia Weichgrebe [1] and Gerhard Zotz [1,2]

1 Functional Ecology Group, Institute of Biology and Environmental Sciences, Carl von Ossietzky University Oldenburg, P.O. Box 5634, 26046 Oldenburg, Germany; flora.letizia.weichgrebe@uol.de (L.W.); gerhard.zotz@uol.de (G.Z.)
2 Smithsonian Tropical Research Institute, Apartado, Balboa, Ancon, Panama City 0843-03092, Panama
* Correspondence: helena.einzmann@uol.de

**Abstract:** As climate change leads to increasing temperatures, tropical dry seasons are expected to become more severe. An overall intensification of drought events may strongly affect vascular epiphytes. Especially at the community level, the response of epiphytes to intense drought events is still poorly understood. Therefore, the severe El Niño event of 2015/16 was used to assess the impact of prolonged drought on an epiphyte community on *Annona glabra* host trees, around Barro Colorado Island. Prior census data from 2002 and 2015 served as a reference for background community dynamics. Net species changes and net population changes at the species level were determined for both periods. While the total abundance of the community almost doubled during the 13 years of the reference period, individual numbers decreased by c. 17% within the year of the El Niño event. Overall, the El Niño event strongly affected the epiphyte community and led to a strong decrease in epiphyte numbers and species. These findings contrast with most previous studies in tropical lowlands that found epiphyte populations to be rather resistant to similarly severe drought events.

**Keywords:** Barro Colorado Island; climate change; drought; El Niño; epiphyte census; tropical lowlands

## 1. Introduction

The El Niño Southern Oscillation (ENSO) phenomenon contributes substantially to interannual variation in climate, particularly in tropical and subtropical countries [1]. In many regions El Niño events are associated with severe droughts, which frequently [2–4], but not universally [5,6], lead to a considerable increase in tree mortality. Structurally dependent plants make up a large fraction of tropical plants and especially woody climbers represent a life form that has attracted considerable attention, particularly during the last two decades [7–9]. Lianas generally seem to suffer much less from drought than co-occurring trees and even benefit from damaged trees due to increased canopy openness and lower below-ground competition [8]. However, other structurally dependent life forms like epiphytes and non-woody climbers or ground herbs have been studied very little with respect to their susceptibility to drought, associated with El Niño events. Such events are predicted to increase in intensity and frequency [10] and to understand how this will affect tropical forests it is essential to study all the different life forms of such a complex system.

Epiphytes have been called "particularly" vulnerable to climate change [11,12], a notion contested by Zotz and Bader [13], at least for epiphytes in seasonal lowland forests. Mostly growing directly on bark, these lowland epiphytes already cope successfully with normal seasonal droughts and may thus suffer less, rather than more, compared to ground-rooted flora during the extended droughts during El Niño events. A number of observations back this notion. Two demographic studies with an epiphytic orchid [14] and an epiphytic bromeliad species [15] on Barro Colorado Island, Panama, with seven annual censuses, found very little effects of the severe 1997/98 El Niño event. Population growth rates did

not correlate with precipitation in either case; even the impact on growth and survival of the smallest and thus most vulnerable individuals was very limited. This contrasts strongly with the severe impact of El Niño droughts on trees in the same forest [16]. Similarly, Einzmann et al. [17] reported a 60% increase in the number of individuals of an epiphyte community from 1994 to 2002, in spite of the extreme 1997–1998 El Niño drought, and a 30% increase from 2002 to 2015, in spite of three consecutive years with below-average precipitation. Even epiphytes in a Costa Rican montane cloud forest, which should be particularly susceptible to rare drought events, survived a severe drought related to an El Niño event [18]. To conclude, current evidence does not support the notion that epiphytes are very much affected by El Niño events. However, this conclusion is based on direct evidence for very few species at the individual or population level and only indirect evidence at the community level [17,19,20].

All available long-term data sets with vascular epiphytes [17,19,21–23] suggest that epiphyte communities at the tree or plot level are not saturated, which leads to continuous increases in individual and species numbers over decades. Any negative effects of an El Niño event at the community level may thus have gone unnoticed in long-term studies such as Laube and Zotz [19] or Einzmann, Weichgrebe and Zotz [17] because losses may simply have been compensated by strong increases during subsequent years.

This shortcoming is overcome in the present study by directly studying epiphytes in *Annona glabra* L. before and after the 2015/16 El Niño event. With 173 days, the 2015/16 El Niño event resulted in the third-longest dry season in Panama on record [24]. While the 2015 census was part of a long-term study on community dynamics, with censuses in 1994, 2002 and 2015 [17,19,20], the 2016 census was conducted for the sole purpose of documenting the possible effects of this El Niño event. We quantified its effect by comparing the net mortality of all epiphytes in 145 trees with the annual background mortality during the preceding 2002–2015 period.

## 2. Materials and Methods

The study was conducted in the Barro Colorado Natural Monument (BCMN, field station: 9°09′53.2″ N, 79°50′13.6″ W), located in Lake Gatún, that was formed 1914 with the construction of the Panama Canal in the Republic of Panama. The area is dominated by a semi-deciduous lowland forest that receives, on average, 2660 mm annual precipitation with a pronounced dry season of four months [25]. *Annona glabra* trees are only found at the lakeshore. In 1994, 2002 and 2015 [17,19,20], c. 1000 trees were censused to document the long-term dynamics of the epiphyte community. While each tree was labeled, individual epiphytes were not marked. Twelve months after the 2015 census, in September/October 2016, a subset of 145 trees was censused again. Sample trees were selected before fieldwork started, guided by the goal to maximize epiphyte diversity, with preference given to trees hosting many epiphyte species. As *A. glabra* trees are multi-stemmed, we noted the following conditions of each tree and of each stem: (a) present and living, (b) dead or (c) missing. For epiphytes, the number of individuals per epiphyte species and per tree stem was recorded in the year before the El Niño event. In the following year, we noted the number of individuals that were (a) found again, (b) dead, (c) missing or (d) newly found. Only epiphytes that had at least 20% of the maximum size of the respective species were counted [compare 19], as mortality of smaller individuals is high and identification is difficult to impossible. Smaller individuals were simply noted as present, but not included in any analysis. All conspecific epiphyte individuals of a given tree are called a "population," due to the close proximity of stems within one tree compared to the stems of other trees. Species names follow The Plant List [26].

The impact of the El Niño event on the epiphyte community was assessed as the net change in species (species year$^{-1}$ tree$^{-1}$) and individual numbers (individuals year$^{-1}$ tree$^{-1}$), hereafter referred to as the net species change (NSC) and net abundance change (NAC), respectively. These numbers were compared with the average changes during the preceding 13 years, using the census data from the same 145 trees from 2002 by Laube and Zotz [19], and from 2015 by Einzmann, Weichgrebe and Zotz [17]. The precipitation pattern of the El Niño event clearly deviated from that of the reference period. In the reference period, precipitation was slightly higher than the long-term average since 1925, c. 2800 vs. 2660 mm a$^{-1}$ [27], with yearly precipitation being eight times above and seven times below the long-term average. However, the two years leading up to the 2015/16 El Niño event had already seen a precipitation below average with only 77% and 83%, respectively [27].

All statistical analyses were conducted in the program R version 4.0.3 [28], and all plots were created with the package ggplot2 [29]. To test if the dynamics during the El Niño event differed from that of the preceding reference period, paired *t*-tests were performed on NAC and NSC for each of the ten most abundant species. Since we only expected negative or neutral effects of the El Niño event on epiphytes, one-sided *t*-tests were chosen. The tested null hypothesis for each test assumed no difference between the sample means of both periods, the alternative hypothesis stating that the mean of the El Niño period was lower than the mean of the reference period. To compare the number of epiphytes on damaged and undamaged stems, another one-sided *t*-test was conducted, and one on the ratio of missing epiphytes per stem for damaged and undamaged stems.

## 3. Results

In 2015, we found 8074 epiphyte individuals of 52 epiphyte species in 13 families, on the 145 individual trees of *A. glabra* (Table S1). The number of individuals decreased to 6715, while the number of species and families hardly changed after the El Niño event; three species were lost on the surveyed trees within this year and two new species were found (Table S1). This represents a net loss of 17%, which results from 10% of the original individuals found dead, another 14% were completely missing and 8% of the individuals in 2016 were newly found (Table 1). Of all dead and missing individuals, substrate failure (i.e., branch- or tree fall) was responsible for only 11%. The ten most abundant species made up >90% of the surveyed individuals in 2015 and 2016, respectively.

**Table 1.** Changes in the number of epiphyte individuals before and after the El Niño event. Percentages are based on the total number of individuals found before the El Niño event. The last row gives the net change from 2015 to 2016.

| Name | Individuals | % |
|---|---:|---:|
| Pre El Niño | 8074 | |
| Post El Niño (still present) | 6085 | 75 |
| **Changes 2015 to 2016** | | |
| Dead | 841 | −10.4 |
| Missing | 1148 | −14.2 |
| Newly found | 630 | +7.8 |
| **Net change** | | −16.8 |

Over the reference period (between 2002 and 2015), the number of epiphytes on the 145 trees increased by 88% (from 4298 to 8074), which represents an average annual net growth of the epiphyte community of 5%. However, this number underestimates the true dynamics. Based on the net changes per tree stem and epiphyte species, at least 52% of the initially recorded individuals were not present any more after 13 years, and at least 74% of all the individuals surveyed in 2015 were newly recorded.

During the El Niño event epiphytes were performing considerably worse than during the reference period. For one, the annual NSC was significantly lower in the El Niño period (Figure 1). While the number of epiphyte species per tree increased during the preceding 13 years with an annual rate of $0.2 \pm 0.2$ species year$^{-1}$ tree$^{-1}$ (mean $\pm$ SD, $n = 145$), species numbers decreased during the El Niño event ($-0.2 \pm 0.9$ species year$^{-1}$ tree$^{-1}$, $n = 145$).

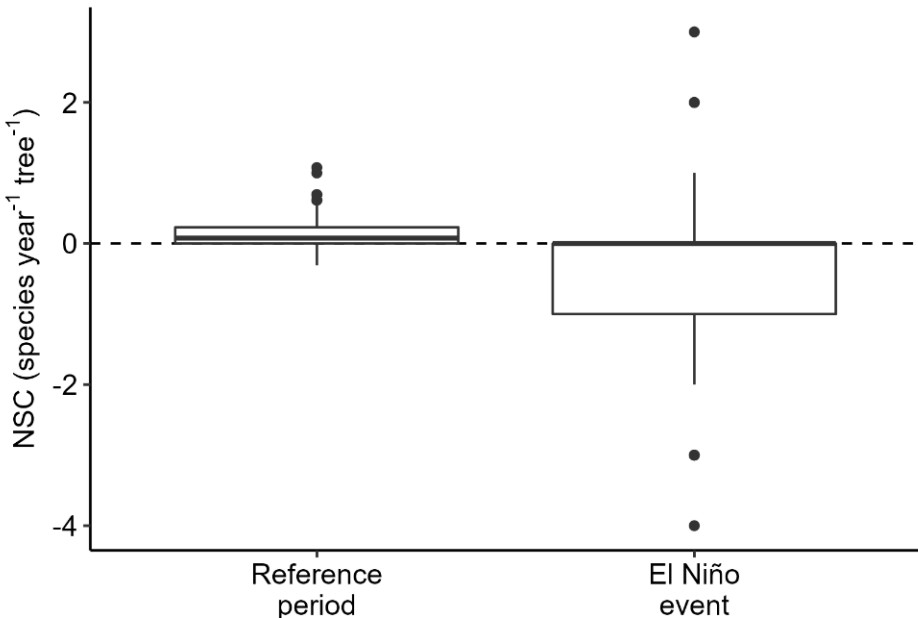

**Figure 1.** Net change in species numbers per year and tree individual (NSC, species year$^{-1}$ tree$^{-1}$) for the reference period and the 12 months with the El Niño event. The change in the number of epiphyte species per tree was significantly lower during the El Niño event ($t = 5.0$, $df = 144$, $p < 0.001$, estimated mean of the difference: 0.4 species year$^{-1}$ tree$^{-1}$).

Similarly, most of the epiphyte species showed a net decrease in population size during the El Niño event compared to the typical net gain during the reference period. The ten most abundant species gained, on average, $0.5 \pm 1.3$ individuals year$^{-1}$ tree$^{-1}$ (mean $\pm$ SD, $n = 145$) during the reference period, but lost $2.5 \pm 7.8$ individuals year$^{-1}$ tree$^{-1}$ during the El Niño period. For seven of the ten most abundant species, this difference was significant (Figure 2). In the extreme case of *Niphidium crassifolium* losses amounted to, on average, $4.4 \pm 8.8$ individuals year$^{-1}$ tree$^{-1}$ (mean $\pm$ SD, $n = 145$) during the El Niño event. Since the NAC of the reference period was averaged over 13 years, its variation was much lower than of the NAC during the El Niño event. Some extreme changes in individuals were observed after the El Niño event in relatively rare species, which resulted in very high standard deviations: *Epidendrum nocturnum*, e.g., lost 57 individuals on a single tree, while the net loss was only 72 individuals. With few exceptions (e.g., *Specklinia brighamii*), rarer species also performed worse during the El Niño event than during the reference period.

Eight of the ten most abundant species decreased markedly in abundance, i.e., with losses between 13% and 28%. The extreme case is *N. crassifolium* (28%) with a loss of 385 individuals from its 1357 individuals before the El Niño event. The most abundant epiphyte species in *A. glabra*, *Caularthron bilamellatum*, decreased by about 19% (from 1910 to 1545 individuals). Two of the ten most abundant species, *Polystachya foliosa* and *Tillandsia subulifera*, showed hardly any change in numbers (0% and −1%, respectively). Some of the rarer species, on the other hand, even increased substantially in abundance. Examples are *Specklinia brighamii* (400% increase, 8 to 40 individuals) or *Campyloneurum phyllitidis* (56%, 18 to 28).



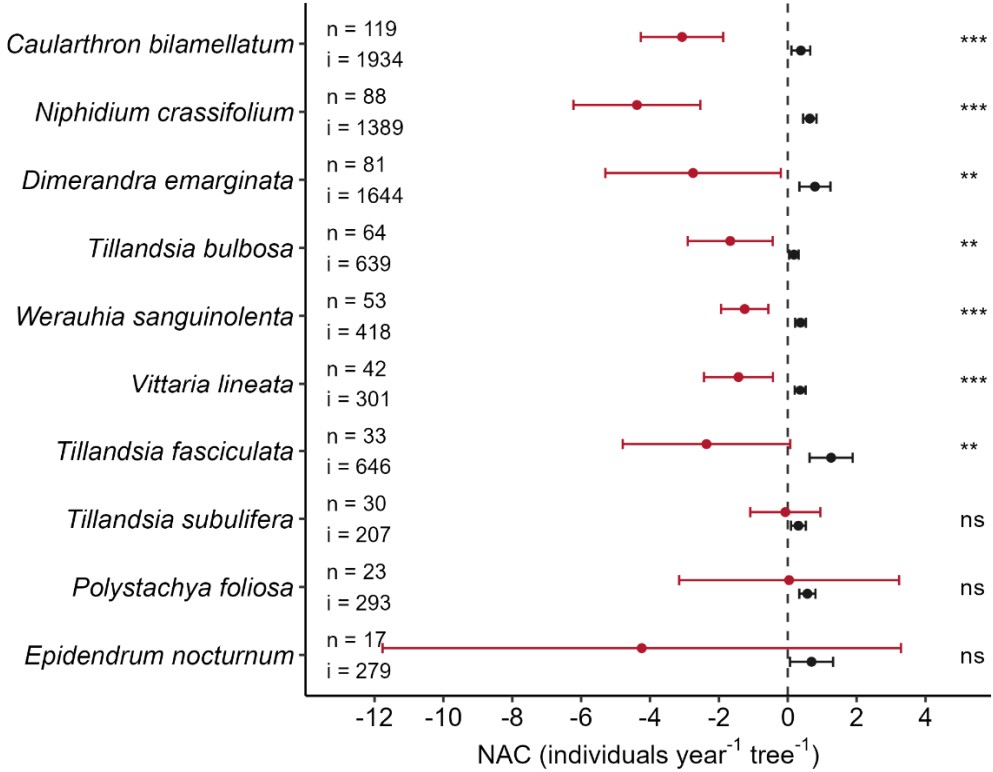

**Figure 2.** Net abundance change (NAC, individuals year$^{-1}$ tree$^{-1}$) of the ten most abundant epiphyte species for the reference period (black) and for the 12 months with the El Niño event (red). Dots give the mean NAC and ranges the 95% confidence interval of the mean NAC for each period and species. The number of populations per species is given as '*n*' (= number of occupied trees) and the total number of individuals during the El Niño period as '*i*'. Totals refer to the total number of observed occupied trees/individuals in 2016, e.g., *Tillandsia subulifera* occurred on 29 trees in 2015, but on only 25 trees in 2016, one of which was not occupied in 2015, thus the total is *n* = 30. For each species, one-sided paired *t*-tests were conducted for the differences in means between the reference and the El Niño period. Significance levels are given as asterisks ($p < 0.001$: ***; $p < 0.01$: **; $p \geq 0.05$: not significant, ns).

Overall, the 145 surveyed *A. glabra* trees were made up of 577 stems in 2015. Of these, 19% were found to be dead, damaged or broken and 4% were entirely missing after the El Niño event. While 40% of all trees had at least some dead, broken or missing stems, only one tree was completely missing. There was no difference in the number of epiphyte individuals on damaged and undamaged stems ($t_{Welch\ Two\ Sample} = 0.8$, $df = 171$, $p = 0.4$), but epiphytes on damaged stems were significantly more often damaged or dead themselves (A: $t_{Welch\ Two\ Sample} = -8.4$, $df = 152$, $p < 0.0001$). On partly broken or dead stems, 42% of the epiphytes were either dead or missing, affecting 10% of all epiphyte individuals.

## 4. Discussion

The total number of epiphytes growing on the 145 *A. glabra* trees *decreased* by 17% during the severe El Niño event of 2015/16, compared to an average annual *increase* of 5% during the preceding reference period of 13 years. Assuming that there is no density dependence in the annual rate of increase, it would thus take more than three years to compensate the losses during this El Niño year.

The epiphyte assemblages in the studied 145 *A. glabra* trees showed, on average, a decrease in species and individual numbers during the El Niño event. At the species level, the most abundant epiphyte taxa suffered disproportionate losses of up to 28% (Figure 2). Our long-term data do not indicate that such a disproportionate susceptibility of common species to El Niño drought would lead to shifts in the rank order of the species growing in

*A. glabra*. Over a period of 21 years, with one major El Niño event, several minor anomalies and two relatively dry years with below-average precipitation (77% and 83%) immediately before the 2015 census, we could only detect a single replacement among the 10 most abundant species and just four replacements among the 20 most abundant taxa [17]. With one exception, there was no change in the rank order of the ten most abundant species during the El Niño either (Table S1).

A surprising finding was the substantial mortality in the tank bromeliad *Werauhia sanguinolenta* during the El Niño period (Figure 2). A demographic study with this species covered the 1997/98 El Niño event, which was even more severe than the one in 2015/16, but that year did not stand out compared to the remaining ones—no demographic process correlated with annual precipitation [15], and population growth rates were invariably positive during the 7 years of that study. The host tree species was identical in both studies, so a possible host-specific effect cannot explain this discrepancy.

This and other relevant studies with vascular epiphytes all indicated a considerable tolerance to El Niño-related drought [14,18,19]. The findings presented here clearly are at odds with this notion. However, studies with other life forms have not always reported consistent results either. For example, while most studies with trees reported substantial increases in mortality [3,7] or reduced growth [30] during El Niño events, others found very little effect, or no increase in mortality, and even a stimulation of growth [6,16]. Most studies with lianas reported a stimulation of growth during dry seasons and during El Niño events, although lianas may suffer more than trees during prolonged drought [7]. The much more limited evidence for epiphytes precludes any generalization about the susceptibility of this life form to El Niño events.

In general, epiphytes at the community level are more affected by drought than any other life form. In their classic study, Gentry and Dodson [31] showed that epiphyte species richness and abundance change much more from dry to moist to wet forest than richness and abundance of co-occurring trees, climbers or shrubs. However, this statement does not mean that epiphytes that do grow in seasonal and drier habitats are particularly vulnerable to drought. To the contrary, while species in very wet systems like montane cloud forests are, indeed, strongly affected by even moderate dry spells [32], species from drier forests have developed impressive capabilities to tolerate drought [33–35].

Individual species differed strongly in their response to the El Niño event. Most conspicuously, the fern *N. crassifolium* showed the most negative response (Figure 2), while the similarly common orchid, *Dimerandra emarginata*, was apparently not affected. Physiological studies of both species, which demonstrated a pronounced tolerance to drought in both cases [34,36,37], do not provide an explanation for these differences.

A considerable proportion of the 52 species of the present study (54%) showed a decrease in abundance, although about a fifth of all species increased in abundance. The notion that species with CAM may be less affected by such a drought event than C3 species has to be dismissed: there was no difference in abundance change between CAM and C3 species in the El Niño year ($\chi^2$ = 19.8, $df$ = 22, $p$ = 0.6, data not shown) nor in the reference period ($\chi^2$ = 40.2, $df$ = 34, $p$ = 0.2, $df$ = 35, $p$ = 0.2, data not shown). The relatively exposed habitat in the *A. glabra* crowns must already filter for epiphyte species capable of coping with long periods of drought.

As vascular epiphyte communities have been shown to be highly dynamic systems [19,23,38], Zuleta et al. [39] argued that epiphyte communities are controlled by a relatively high rate of mortality (7.5% year$^{-1}$). Apart from the direct mortality due to drought, several studies have shown epiphytes are often even more affected by the dynamics of the host [15,19,35,39–42]. Zotz, Laube and Schmidt [15], for example, found that substrate failure (i.e., tree and stem mortality) was the main cause of death of medium-sized individuals in a demographic study with the epiphytic bromeliad *W. sanguinolenta* on *A. glabra*. Similarly, Laube and Zotz [19] considered epiphyte extinctions (of adult individuals) in the *A. glabra* epiphyte community to be mainly related to substrate failure. However, our results suggest that substrate failure only accounted for a small fraction of dead or

missing epiphytes during the El Niño event. Nevertheless, the observed rate of epiphyte mortality due to substrate failure (11% year$^{-1}$) is somewhat higher than rates described by Hietz [40], 7% year$^{-1}$, and by Zuleta, Benavides, López-Rios and Duque [39], 5.6% year$^{-1}$, but considerably lower than in a study of *W. sanguinolenta* growing on *A. glabra* close to BCI, 17–83% year$^{-1}$ [43]. Whether the El Niño event affected epiphytes indirectly via increased mortality of the host tree remains unclear. Although there is no data on the typical dynamics of the *A. glabra* trees in this area, any possible impact on epiphytes during the El Niño event seems to be small. Our findings do not support the suggestion of Zotz and Schmidt [14] that tree mortality could increase due to climate change and subsequently lead to a higher mortality among epiphytes.

## 5. Conclusions

In conclusion, the epiphyte community in *A. glabra* was strongly affected by the prolonged dry season of the El Niño event in 2015/16. Although Einzmann, Weichgrebe and Zotz [17] showed that increases in years with normal climatic conditions have led to a continuous long-term increase in species numbers and individual abundance over one century in this system, the predicted increase in extreme drought events with climate change could substantially change the long-term dynamics of this and other epiphyte communities.

**Supplementary Materials:** The following supporting information can be downloaded at: https://www.mdpi.com/article/10.3390/d14050325/s1, Table S1: Species table.

**Author Contributions:** Conceptualization, G.Z.; formal analysis, L.W. and H.J.R.E.; funding acquisition, G.Z.; investigation, L.W.; data curation, H.J.R.E.; writing—original draft preparation, L.W.; writing—review and editing, H.J.R.E. and G.Z. All authors have read and agreed to the published version of the manuscript.

**Funding:** Funding by the ESP-Program of the Smithsonian Tropical Research Institute is acknowledged (funding number: 34ESP40).

**Institutional Review Board Statement:** Not applicable.

**Informed Consent Statement:** Not applicable.

**Data Availability Statement:** The data presented in this study is archived with the Figshare data repository, available from: https://doi.org/10.25573/data.19629924.

**Acknowledgments:** The help of Anaïs Bonnefond, Montpellier, France during the 2015 census is acknowledged, as are ESP funds from the Smithsonian Tropical Research Institute.

**Conflicts of Interest:** The authors declare no conflict of interest.

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
