# Peer review of "The Impact of a Severe El Niño Event on Vascular Epiphytes in Lowland Panama"

_diversity, doi:10.3390/d14050325_

Round 1
Reviewer 1 Report
It is necessary to review how scientific names are cited. The first time the name appear in the text they MUST be complete, that is, the name of the genus and that of the specific epithet. Afterwards it is enough to cite them with the initial of the generic name and the specific epithet. On the other hand, the authors of the names MUST ALWAYS be indicated the first time it appears in the text, or failing that, in all the names in table S1. The name of the authors represents the citation to their work and MUST be respected.

Author Response
It is necessary to review how scientific names are cited. The first time the name appear in the text they MUST be complete, that is, the name of the genus and that of the specific epithet. Afterwards it is enough to cite them with the initial of the generic name and the specific epithet. On the other hand, the authors of the names MUST ALWAYS be indicated the first time it appears in the text, or failing that, in all the names in table S1. The name of the authors represents the citation to their work and MUST be respected.
Response: The author guideline does not state any preference on how to present species names. We included the author names to the epiphyte species in the supplementary table and give the author name of Annona glabra L. at the first mentioning after the abstract (L69). Following your suggestion, in the text we abbreviated all genus names to their first letter after their first mentioning, except at the start of a sentence.
From the comments in the text:
This indicates only one point. Is that point in the center of the reserve? It is necessary to indicate two data in longitude and two data in latitude
Response: Two location points would not give a more precise location of the trees either. We now give the location of the field station and indicate this.
Reviewer 2 Report
The manuscript is well written and structured, some minor suggestions in the methods and results sections are included; it is also important the correct citation of the scientific names. In the supplementary material the scientific names must be completed with the cite of authors and probably consultation would be easier if they were organized by family

Author Response
The manuscript is well written and structured, some minor suggestions in the methods and results sections are included; it is also important the correct citation of the scientific names. In the supplementary material the scientific names must be completed with the cite of authors and probably consultation would be easier if they were organized by family
Response: The author guideline does not state any preference on how to present species names. We included author names and families in the supplementary table. However, we prefer the table to present the ranking of the abundance than to order it by family.
From the comments in the text:
It could be convenient to include a location map of the study area
Response: The study builds on several other studies of this system, thus a map can be found in e.g. Zotz et al. 1999 Journal of Biogeography or Einzmann et al. 2021 Journal of Ecology. In addition, the study site is the most intensively studied tropical forest in the world and thus well known. Therefore, we decided not to include a map here. We can provide one, if the journal team thinks it is necessary.
It is necessary to put the author(s) in all scientific names the first time they appear in the text
Response: The author guideline does not state any preference on how to present species names. Since we now included author names to the epiphyte species in the supplementary material and another reviewer also deemed this sufficient, we prefer to keep the text clearer by omitting the author names in the text.
It would be important to explain why this percentage […epiphytes that had at least 20 % of the maximum size…]
Response: We included an explanation (L93-94).
It would be interesting to have this information available in an appendix […epiphyte individuals of 52 epiphyte species in 13 families…]
Response: This information is available in the supplementary material. We included a reference now (L121).
It is important to mention these species […three species were lost on the surveyed trees within this year and two new species were found…]
Response: This information is available in the supplementary material. We included a reference now (L124). It is not essential for the understanding to mention the species here.
Is a different paragraph, the name of the genus must be completed / Put the complete name of the ganus
Response: Another reviewer was of different opinion regarding the first point and this neither is a general rule nor specified in the author guidelines. We made sure that we implied a consistent rule now.
Reviewer 3 Report
Please refer the attached document for detail comments.

Author Response
Comments to authors,
The study compared a vascular epiphyte community (species and abundance net changes) on 145 trees in Panama before and after a severe El Niño event during 2015/16. The project was actually across 13 years beginning from 2002 of the local epiphyte censuses. They found despite extreme El Niño in 1997/98, local epiphytes were doubled in abundance from 2002 to 2015, while individual and species numbers were decreased c. 17% after 15/16 El Niño event. Seven of ten most abundant epiphytic species showed significantly loss from El Niño impacts. Neither physiological characteristics of epiphytes nor the host tree mortality rate accounted for the negative responses of epiphytes to drought stress of the El Niño event. The study provided first-hand/ long-term investigations on neotropical lowland forests, and results suggested different ecologic groups and ecosystems may receive various, even contrast, influences from El Niño, which is interesting for climate change science.
Response: Thank you for the positive reception.
Minor comments (according to the page and line number in the PDF file):
Line 27: Barro Colorado Island, El Nino, repeated in abstract and title
Response: Author instructions don’t specify if repetitions should be avoided, depending on the search algorithm a repetition of the terms in the keywords can be useful.
Line 38: “below-ground” replaced as understory
Response: No, this refers to below-ground competition for resources.
Line 41: “how this will affect tropical forests it is essential to…”
Response: Actually, the sense of the sentence gets lost/changed if “it” is deleted – it needs to stay.
Line92~93: Do you mean the epiphyte that compared with study [19] and over 20% of the pervious individual numbers was counted?
Response: The comparison refers to the method of including only individuals of at least 20 % of the maximum size of a species. We now provide a brief explanation of the rationale (L93-94).
Line 103: Not clear of ”eight & seven times” description.
Response: Rephrased to increase clarity (L103-107).
Line 124: Define “substrate failure” for the first time in the manuscript.
Response: Done (L126-127)
Line232~235: Studies pointed out ferns have various stress tolerance during life cycles. Would it be possible to explain its negative response?
Response: Since we lack any information on the gametophyte stage, we cannot comment on this. Anyway, such detailed analysis is actually beyond the scope of our study, but could be the subject of follow-up studies.
Reviewer 4 Report
I read the manuscript and it is written well. Yet, I have some concerns and I could not understand in some places. So, please see my comments below:
Authors have mentioned that there is decline in epiphyte species number and abundance due to EL Nino. However, I am not sure if there were any natural disasters like storm, etc. This information is missing and how can we be sure decrease in number is simply because of El Nino. I suggest authors to clarify this missing link. In the nature decrease or increase could also be due to reproductive rates of the plant species. What is the means of reproduction or propagation?
Abstract
Line 22: ‘Similarly strong changes in species numbers and population sizes occurred during the El Niño event..’ replace ‘changes’ by ‘decrease’ or ‘reduction’.
Conclusion could perhaps be as: Our finding showed that increase in temperature negatively affect plant growth and survival and our study can be as base line information for further climate change related studies, mainly epiphytes in general.
Introduction: try to include some studies that see the effects of drought on epiphytic plant species. I know there are some but try to dig in literature and show that epiphytes in Panama are not studied in this respect. Now it looks as if you want to study and just carried it out.
Method looks fine.
Results and discussion: clear enough to understand them.
Conclusion: as mentioned above.
Probably authors should write a paragraph on implication of this study.
Author Response
I read the manuscript and it is written well. Yet, I have some concerns and I could not understand in some places. So, please see my comments below:
Authors have mentioned that there is decline in epiphyte species number and abundance due to EL Nino. However, I am not sure if there were any natural disasters like storm, etc. This information is missing and how can we be sure decrease in number is simply because of El Nino. I suggest authors to clarify this missing link. In the nature decrease or increase could also be due to reproductive rates of the plant species. What is the means of reproduction or propagation?
Response: El Niño events typically encompass drought or heavier rainfall (depending on the region, in the study region that is drought) and intensification of storms. El Niño therefore is not only drought. Since for epiphytes the principal abiotic challenge appears to be drought, we focussed on this aspect in our discussion throughout the manuscript. We don’t see that any other natural disaster apart from more localized phenomena should have an impact on the Annona glabra trees around BCI and its epiphyte assemblages.
Any impact of El Niño on reproduction would not be detected in our study. Early growth is so slow that any new recruit would not appear in our data set. We document the short-term effect immediately visible after the event on already established plants.
Abstract
Line 22: ‘Similarly strong changes in species numbers and population sizes occurred during the El Niño event..’ replace ‘changes’ by ‘decrease’ or ‘reduction’.
Response: The replacement would be wrong. However, because the sentence appears to be confusing and is not essential here, we deleted it entirely.
Conclusion could perhaps be as: Our finding showed that increase in temperature negatively affect plant growth and survival and our study can be as base line information for further climate change related studies, mainly epiphytes in general.
Response: We disagree. This is not about temperature. The main problem with El Niño in Panama is drought and the potential of drought to affect epiphytes is well established.
Introduction: try to include some studies that see the effects of drought on epiphytic plant species. I know there are some but try to dig in literature and show that epiphytes in Panama are not studied in this respect. Now it looks as if you want to study and just carried it out.
Response: We are confused – an entire paragraph (L43-62) is dedicated to the effect of drought on epiphytes with examples mostly from Panama. We feel that this paragraph is giving exactly the context this reviewer is asking for.
Method looks fine.
Results and discussion: clear enough to understand them.
Conclusion: as mentioned above.
Probably authors should write a paragraph on implication of this study.
Response: We feel that we already did this.